# Identification of Potential Inhibitors of MurD Enzyme of *Staphylococcus aureus* from a Marine Natural Product Library

**DOI:** 10.3390/molecules26216426

**Published:** 2021-10-25

**Authors:** Xiaoqi Zheng, Tongyu Zheng, Yinglin Liao, Lianxiang Luo

**Affiliations:** 1The First Clinical College, Guangdong Medical University, Zhanjiang 524023, China; zxqzxq@gdmu.edu.cn (X.Z.); dzheng@gdmu.edu.cn (T.Z.); lyl212608@gdmu.edu.cn (Y.L.); 2The Marine Biomedical Research Institute, Guangdong Medical University, Zhanjiang 524023, China; 3Southern Marine Science and Engineering Guangdong Laboratory (Zhanjiang), Zhanjiang 524023, China; 4The Marine Biomedical Research Institute of Guangdong Zhanjiang, Zhanjiang 524023, China

**Keywords:** *S. aureus* MurD enzyme, virtual screening, homology modelling, ADME, molecular dynamics

## Abstract

*Staphylococcus aureus* is an opportunistic pathogen that can cause fatal bacterial infections. MurD catalyzes the formation of peptide bond between UDP-*N*-acetylehyl-l-alanine and d-glutamic acid, which plays an important role in the synthesis of peptidoglycan and the formation of cell wall by *S. aureus*. Because *S. aureus* is resistant to most existing antibiotics, it is necessary to develop new inhibitors. In this study, Schrodinger 11.5 Prime homology modeling was selected to prepare the protein model of MurD enzyme, and its structure was optimized. We used a virtual screening program and similarity screening to screen 47163 compounds from three marine natural product libraries to explore new inhibitors of *S. aureus*. ADME provides analysis of the physicochemical properties of the best performing compounds during the screening process. To determine the stability of the docking effect, a 100 ns molecular dynamics was performed to verify how tightly the compound was bound to the protein. By docking analysis and molecular dynamics analysis, both 46604 and 46608 have strong interaction with the docking pocket, have good pharmacological properties, and maintain stable conformation with the target protein, so they have a chance to become drugs for *S. aureus*. Through virtual screening, similarity screening, ADME study and molecular dynamics simulation, 46604 and 46608 were selected as potential drug candidates for *S. aureus*.

## 1. Introduction

Bacterial infections have become a common case worldwide, and the development of antibiotics has been hampered by bacterial resistance. *S. aureus* belongs to the genus Staphylococcus, which can cause a variety of serious infections. According to the sensitivity to antibiotics, *S. aureus* can be divided into *methicillin sensitive Staphylococcus aureus* (MSSA) and *methicillin resistant Staphylococcus aureus* (MRSA). In recent decades, due to the evolution of bacteria and the abuse of antibiotics, the drug resistance of *S. aureus* has gradually increased, the infection rate of MRSA has increased all over the world, and the clinical anti-infection treatment of MRSA has become more difficult [1]. *S. aureus* is the most common pathogen causing septic infections in humans, which can lead to localized septic infections, pneumonia, pseudomembranous enteritis, pericarditis, and even systemic infections such as sepsis. It is a bacterial infection with a high mortality rate [2]. The virulence of *S. aureus* mainly depends on its toxin and invasive enzyme. Among them, the harm of enterotoxin has become a worldwide health problem. The infections caused by *S. aureus* are second only to that caused by *E. coli*, which was at a record high. Peptidoglycan is an essential part of the *S. aureus* cell envelope, and its presence is associated with cell integrity. The formation of peptidoglycan depends on the ATP-dependent bacterial Mur ligase (MurC-F) to catalyze the addition of l-alanine (l-Ala), d-glutamic acid (d-Glu) and meso-2,6-diaminoheptanoic acid (meso DAP) to Gram-positive or l-lysine. In Gram-positive bacteria, d-alanine-d-alanine (d-Ala-d-Ala) dipeptide is transformed into UDP-*N*-acetyl muramic to form UDP muramic pentapeptide [3]. d-glutamate ligase (MurD) catalyzes ATP-dependent phosphorylation of UMA carboxylic acid. The resulting acyl-phosphate intermediate is then attacked by the d-Glu introduced into the amino group, thus forming a high-energy intermediate tetrahedral, which finally collapses into an amide product, UDP-*N*-acetyl-muramyl-l-alanine-d-glutamate (UMAG), and inorganic phosphate [4,5]. The emergence of drug-resistant *S. aureus* and the decline in the production of new antibiotic treatment schemes in the pharmaceutical industry are increasingly serious global healthcare problems. The drug R&D industry has not stopped the R&D of *S. aureus* inhibitors, but there are still no breakthrough results. The coadministration of pump inhibitor (EPI) and antibiotics as pump substrate could increase the intracellular drug level, thus bringing new curative effects to the existing anti-staphylococcal drugs. As the available treatment options are limited, *S. aureus* strains have emerged against widely distributed antibiotics (including β-Lactams, aminoglycosides, macrolides, fluoroquinolones (FQs), chloramphenicol, sulfonamides, streptomycin and tetracycline) which are resistant, and the existence of resistance is an important link to hinder drug research and development [6]. At present, the inhibitors against *S. aureus* mainly include non-phosphorylated and phosphorylated hydroxyethylamine [7], d-glutamate and substituted naphthalene-*N*-sulfonyl-d-glutamate analogues [8,9,10]. Several hypophosphonates and phosphonate inhibitors have been developed, such as tetrahedral transition state analogues of *E. coli* MurD and *N*-acetyl cell wall acid analogues, but no promising results have been obtained [11].

Almost all life forms in the marine environment, such as algae, sponges, corals and sea squirts, have been investigated for the content of their natural products. Many structurally and pharmacologically vital substances have been isolated with new antibacterial, antitumor and anti-inflammatory properties. With the development of terrestrial antibiotics entering the bottleneck period, marine drugs are gradually becoming the source of new drugs [12,13,14]. In this study, we have established a three-dimensional protein structure of MurD enzyme by homology model in the absence of a protein model of *Staphylococcus aureus* MurD enzyme, and then its structure was optimized by GROMACS 2019.1. The SiteMap module of Maestro 11.5 was used to predict the location of protein active amino acids and establish docking pockets. After virtual screening, 2010 compounds with the top 10% docking score were obtained. The results were screened based on 3D structure by using the inhibitors of existing studies. Finally, 20 compounds with the best similarity score were selected to evaluate their drug properties and calculate their binding energy. Then, we ran simulations of the most promising compounds, 46604 and 46608, in complex with protein formation by GROMACS 2019.1.

## 2. Materials and Methods

### 2.1. Homology Modelling and Protein Preparation

The protein model of MurD enzyme of *S. aureus* was established by Maestro 11.5 prime module (Schrödinger, Shanghai, China). From the UniProt sequence database (http://www.uniprot.org/uniprot/, accessed on 15 September 2021), the complete amino acid sequence of the target protein MurD of *S. aureus* was retrieved in FASTA format, and Q6GHQ2 was finally selected as the target sequence. The sequence was then imported into the structure prediction wizard. Because the database in Maestro 11.5 was incomplete, NCBI BLAST (https://blast.ncbi.nlm.nih.gov/Blast.cgi, accessed on 15 September 2021) was used to sequence alignment, which was performed based on 46.85%, 0% gap and 66% sequence identity. The crystal structure of the apo form of MurD ligase from *Streptococcus agalactiae* with 1.5 Å resolution (PDB: 3lk7) was selected as the template [15,16]. The target protein (PDB: 3LK7) was downloaded from the PDB database (https://www1.rcsb.org/, accessed on 15 September 2021) and imported into the Prime module for sequence alignment.

In addition, the GROMACS 2019.1 software package (https://manual.gromacs.org/documentation/2019.1/download.html, accessed on 15 September 2021) was used for molecular dynamics simulation optimization, and the force field gromos54a7 and SPC model were used to establish the initial system. In order to ensure the total charge neutrality of the simulation system, a corresponding number of sodium ions were added to the system to generate a solvent box of appropriate size. The periodic boundary condition was applied to the system. First, the energy of 50,000 steps of the whole system is minimized at a temperature of 300 K. Under position constraints, NVT ensemble (constant particle number, volume and temperature) and NPT ensemble (constant particle number, pressure and temperature), the equilibrium of protein and solvent was realized by MD simulation. Finally, the optimal model was selected according to the equilibrium.

### 2.2. Reactive Pocket Preparation and Ligand Treatment

To obtain the binding pockets of potential ligands, the Maestro 11.5 SiteMap module was used to predict the binding sites of the protein structures. SiteMap is a useful tool to accelerate the drug discovery process, complementing techniques such as docking and structure-based lead optimization [17]. The SiteMap tool was used to identify the binding pocket of MurD enzyme protein. Sites with scores higher than 1 were selected for grid generation and docking studies using glide grids (Table 1). The receptor grid generation tool in Maestro 11.5 was used to generate a grid containing ligands for the sites with the highest score. The van der Waals (VDW) radius scale factor of nonpolar receptor atoms was 1.0, and the partial charge cutoff value is 0.25. The grid is set to 20 Å × 20 Å × 20 Å, and the grid is generated by selecting and entering specific residues. MurD’s receptor grid is generated by specifying binding (active) site residues that are identified by the SiteMap tool [18]. Ligands are from the seaweed metabolite database (SWMD) (http://www.swmd.co.in, accessed on 10 September 2021), Marine Natural Product Database (CMNPD) (https://www.cmnpd.org/, accessed on 10 September 2021), Marine Natural Products (MNP) (http://docking.umh.es/, accessed on 10 September 2021). A total of 47,163 marine small molecule compounds were imported into Maestro 11.5, and the small molecules were processed with the LigPrep module. The opls3 force field and Epik were used to generate ligand ionization states in the pH range of 7.0 ± 2.0, generate low-energy stereoisomers for each ligand, retain stereoisomers with correct chiral low-energy 3D structure, and produce more than 80,000 small molecules with different conformations.

### 2.3. Receptor-Based Virtual Screening

In order to screen out more promising small molecules, the Qikprop module was used to implement Lipinski’s five rules to screen compounds. Violation of such rules will prevent the compounds from further VSW. Lipinski’s five principles include that the relative molecular weight of the compound is less than 500, the number of hydrogen bond donors (including hydroxyl, amino, etc.) is not more than 5, the number of hydrogen bond receptors in the compound is not more than 10, the logarithm of the lipid water partition coefficient (logP) of the compound is between −2 and 5, and the number of rotatable bonds in the compound is not more than 10. Compounds are more likely to exhibit malabsorption or poor penetration when two or more criteria are not met. Structure-based virtual screening was carried out through the Maestro 11.5 virtual screening workflow module. Virtual screening was carried out in two stages: (a) standard docking (SP) and (b) precision docking (XP). In the selection of output quantity, the compounds with the top 30% and the top 10% of the selection score after XP docking were reserved in SP mode for subsequent steps.

### 2.4. Shape Screening

Shape-based methods for arranging and scoring ligands have been proven valuable in the field of computer-aided drug design. A flexible ligand superposition and virtual screening method based on shape, phase shape, has been proved to quickly generate accurate 3D ligand comparison and effectively enrich active substances in virtual screening. Phase shape depends on the rapid superposition of atomic pair similarity and atomic triples. The scoring function can only contain shape overlap or a combination of shape and atomic/pharmacophore characteristics. Shape screening can be divided into two-dimensional (2D) screening and three-dimensional (3D) screening. The 2D structure screening method is established to obtain more identical parts between paired molecules, and the 3D screening method can be used to enhance the diversity of scaffolds. Shape screening based on 3D structure is mainly to compare the 3D geometry between molecules and score by calculating the overlap between the volumes of rigid spheres. The scoring method can consider only the shape or both the shape and the characteristics of atoms/pharmacophores. In this experiment, we chose to score according to the latter [19]. Considering that *S. aureus* has developed resistance to most antibiotics, the template inhibitor selected in this study came from the redesigned compound D1 in literature [16], and the 2D structure of D1 is shown in Figure 4 below. The 2D structure of the inhibitor was drawn in Maestro 11.5 and converted to a 3D structure. Subsequently, hydrogen atoms were added to the structure in LigPrep, all possible ionization states were generated between pH 5.0 and pH 9.0 using the ion generator module, and the 3D molecular structure was minimized using the opls-2005 force field. A shape base screening was performed for 2010 compounds and inhibitor D1 after precise docking, and normalized shape similarity values were calculated for each molecule conformation relative to ly-1-100; 0 represents the most dissimilar and 1 represents the identical shape. The hit threshold selects shape similarity greater than 0.4, and compounds with similarity greater than 0.4 will be output [20].

### 2.5. ADMET Analysis

Drug formation analysis was completed by using the Discovery Studio 4.5 ADMET module (Dassault, Paris, France) and Maestro 11.5 Qikprop module to check the drug properties of molecules. The ADMET module was based on atomic logP98, polar surface area (PSA), blood–brain barrier (BBB), cytochrome P450 and hepatotoxicity. The blood–brain barrier was divided into different levels, 3 (low: brain–blood ratio less than 0.3:1) and 4 (undefined: outside 99% confidence ellipse). In addition, the polar surface area was calculated by using the method based on the sum of the list surface contributions of polar fragments called topological PSA (TPSA). Compounds with PSA < 140 and logP98 < 5 represent higher cell permeability. ADME was also calculated through Maestro 11.5 Qikprop module, and the calculated characteristics include HERG, CNS activity, H-bond donor, H-bond receptor, log P_O_/_W_, logS, log BB and PMDCK.

### 2.6. Molecular Dynamics

GROMACS 2019.4 ran on a high-performance Linux cluster to determine the behavior of the selected ligands 46604 and 46608 complex with *S. aureus* MurD enzyme protein within 100 ns. We used the Bio2byte Web server (https://www.bio2byte.be/, accessed on 25 September 2021) to generate topology files for the two ligands [21]. From the docking study, the ligand protein complex with the most drug-forming and good docking status was selected as the input file of MD simulation. The force field uses amber99sb-ildn.ff. The complex of the TIP3P water model was surrounded by the dodecahedral-shaped water tank. In order to neutralize the net charge of the system, Na and Cl counter-ions were replaced by water molecules. The steepest descent algorithm with a tolerance of 1000 kJ/mol/nm was used to minimize the energy of the system. The cutoff value of van der Waals was 12 Å, and periodic boundary conditions were specified in all directions. After convergence, NVT ensemble MD simulation was within 100 ps, and then the system passes through NPT within 100 ps under periodic boundary conditions. Berendsen constant pressure and thermostat were used to maintain the temperature and pressure at 300 K and 1 bar for coupling time of τ T = 0.1 ps, τ p = 2 ps. The particle grid Ewald (PME) method was used to calculate the long-range electrostatic interaction. The LINCS algorithm was used to limit the key length. For both ligands, the MD simulation run of 100 ns was repeated twice at constant temperature and pressure, and the average value of the results was reported [22,23,24].

## 3. Results

### 3.1. Homology Modeling and Validation

Since the MurD enzyme catalytic domain is lacking in the protein model, the Maestro 11.5 prime module was used for homology modeling in the study, while the MurD crystal structure of *Streptococcus agalactiae* (PDB: 3lk7) was used as the template. After comparison, it showed that it had 46.85% homology. The generated protein sequence is shown in Figure 1A below. Then, the structure of the protein is optimized by GROMACS 2019.4 to supplement the wrong part in the side chain. The optimized protein model uses PROCHECK (https://saves.mbi.ucla.edu/, accessed on 17 September 2021) to predict the quality of the model, and the online website PROSA (https://prosa.services.came.sbg.ac.at/prosa.php, accessed on 17 September 2021) to assess the quality of the built models. In the Ramachandran plot produced by the PROCHECK website, Figure 1B shows 88.8% residue in the most popular regions, 9.7% residues were found in the allowed areas, 1.2% residues were found in the generous areas, and only 0.2% residue was found outside. Therefore, it is concluded that 99.8% residues were found in the favorable region, while only 0.2% residue was found in the external region, which proves that the quality of the model is good. Figure 1C,D are the superposition diagrams of the modeling protein and the template, respectively. The PROSA test was used to compare the energy standard between the predicted model and known X-ray and NMR structures. The PROSA energy diagram was calculated to check the interaction energy of all residues of the prediction model [25]. The results showed that the z-score of the optimized protein in Figure 2A reached −11.68, while that of the template protein in Figure 2B was −12.81. The PROSA analysis of the model showed that most residues had negative interaction energy, while a few residues showed positive interaction energy. Figure 2C reflects the interaction energy of the MurD enzyme protein, and Figure 2D represents that of the 3l7k protein. In general, the analysis results of PROCHECK and PROSA prove the accuracy of the homologous modeling model; therefore, this protein can be investigated further [26].

### 3.2. Docking Analysis

The Qikprop module used Lipinski’s five rules to predict compounds, while 60,000 compounds that did not violate Lipinski’s RO5 were left in the VSW. The protein was generated by the active site prediction of the sitemap and the receiver grid generation module 20 Å × 20Å × 20 Å. The prepared docking pocket and the screened small molecules were subjected to VSW screening. In the first stage of docking implementation, SP docking (standard docking) was adopted and the compounds with the top 30% scores were retained. In the second stage, XP docking (precision docking) was adopted to retain the compounds with the top 10% scores. Finally, 2010 compounds were selected into the results; 57568 with a docking score of −11.059 was identified as the best docking compound, while 3702 with a docking score of −5.484 was identified as the worst performing compound in the results. The interaction results of the two are shown in Figure 3. The protein–ligand Interaction Profiler (https://plip-tool.biotec.tu-dresden.de/plip-web/plip/index, accessed on 20 September 2021) was used to analyze the complex interactions. In the compound 57568 with the best docking performance, it can be observed that it plays a hydrophobic role with Phe337, the terminal oxygen atom interacts with residues Thr123 and Asp326, and has π-Cation Interaction with residue Arg311 (2D effect diagram of compound 57568 is shown in Figure 3A, and 3D conformation in protein is shown in Figure 3B). Significantly, the salt bridge established between residues Arg311 and Lys328 also plays a key role in ligand–receptor binding. It is clear that the rich type of interactions between compound 57568 and proteins allow for the best docking results between them. In addition, according to the analysis of compound 3702 with the worst performance, the three benzene rings have strong hydrophobic interactions with residues Lys19, Pro79, Asn145, Phe170, Thr330 and Trp425, respectively, and the nitrogen atom and a double-bond carbon atom on the benzene ring form H bonds with residues Lys19 and Ser20, respectively (2D action rendering of compound 3702 is shown in Figure 3C, and 3D conformation in protein is shown in Figure 3D). The single type of interaction between 3702 and the protein accounts for its low docking score. It can be seen from the interaction analysis that the docking results are reliable, and the selected compounds can be further analyzed.

### 3.3. Shape Screening

There were four options in shape similarity screening, from inaccurate to the most accurate. (1) only shape (all atoms are treated the same); (2) QSAR ((hydrophobic, electron absorbing, hydrogen bond donor), negative ions, positive ions and others); (3) elements (each element has different atomic types); and (4) macro model (including more than 150 unique atomic types). Before screening, the default feature definition is applied to identify the location of the following types of pharmacophore sites: aromatic, hydrophobic, H-bond receptor, H-bond donor, negative ion and positive ion. Each site is represented by a sphere with a radius of 2 Å, and as with the atomic typing scheme, the volume overlap fraction is calculated only between sites of the same type. In order to explore the potential new marine small molecule inhibitors based on the molecular shape of D1 (Figure 4), we searched the shape similarity of the best 2010 compounds in the previous virtual screening. We specify that those with similarity scores higher than 0.4 will be output. In the current experiment, we chose to score according to the pharmacophore. In the final search results, 20 results were output. The most similar 46604 scored 0.447, and the less similar 46636 scored 0.402. Previous studies have shown that the designed D1 inhibitor can stably bind to the catalytic pocket of MurD enzyme [16]. MM-GBSA binding energy analysis and docking verification also showed that D1 can stably bind. The MD simulation results showed the importance of the two benzene rings and polar groups in D1. The oxygen atom in the carbonyl group interacts with the skeleton of Val149 and Lys19 residues respectively, and the hydroxyl group on the benzene ring forms a hydrogen bond with the skeleton of Asn145. The -OH of the carboxylic acid group on the benzene ring forms two hydrogen bonds, each hydrogen bond with the main chains of Gly147 and Thr430. The indole ring of the molecule also shows three π–π stacking interactions with Trp425 and one π–π stacking interaction with His48 [16]. Therefore, the discussion will mainly focus on the analysis of functional groups to ensure the accuracy of screening results.

### 3.4. ADME Drug Potency Analysis

In order to predict the drug prospect of compounds more accurately, the Discovery Studio 4.5 ADMET module and Maestro 11.5 Qikprop module were used to analyze the results of the previous stage. The results obtained are consistent with Lipinski’s five rules. Other indicators were used to predict the drug formation of the compounds in the evaluation process. ADMET module provides indicators such as Solubility, PPB Prediction, logP98, BBB Level, PSA (TPSA) for research. At the same time, this module is also used to establish intestinal absorption model, which contains 95% and 99% confidence ellipses on the ADMET_PSA_2D and ADMET_AlogP98 planes, respectively. In addition, 95% of well-absorbed compounds are expected to fall within the 95% ellipse range, and 99% of well-absorbed compounds should fall within the 99% ellipse (Intestinal absorption model is shown in Figure 5). Additionally, the Qikprop module used predictors such as logS, blood–brain partitioning coefficient, Caco-2 cell permeability, apparent MDCK cell permeability, and oil-water partitioning coefficient for drug-like analysis. Typically, a blood–brain partition coefficient between −3.0 and −1.2 usually correspond to a molecularly poor blood–brain barrier permeability. QPPMDCK less than 25 are considered as very poor, while values greater than 500 are considered as excellent. QPlogHERG is the logarithm of blocking the HERG K^+^ channel. Less than −5 indicates a better performance range. Based on Jorgensen’s 3-rule and drug-forming index analysis, 46604 and 46608 finally become the most promising compounds for molecular dynamics verification (Table 2).

### 3.5. Molecular Dynamics

In order to test the stability of *S. aureus* MurD enzyme protein binding with two ligands, 100 ns molecular dynamics simulation was carried out, and its output was analyzed as follows. The simulation results of the two composites are supplemented in the following materials (Figure 6). In the 100 ns simulation, both complexes can converge to the equilibrium state in the last 25 ns. Among them, the 46604 complex converges to the equilibrium state faster than the 46608 complex. After the two complexes reach the equilibrium state, they can continue to the stable state until the end of the simulation. The 46608 complex showed obvious amplitude, experienced obvious conformational changes, and reached a steady state more slowly. RMSD diagrams of ligands in Figure 6A,B were further analyzed; it can be found that 46608 as a whole is relatively stable in this process without too much fluctuation, while 46604 has some fluctuation in the first 75 ns, and finally reaches equilibrium in about 80 ns. The detailed analysis of the root mean square fluctuation (RMSF) of each residue base in Figure 7 is also used to determine the residue fluctuation and flexibility throughout the simulation period. The RMSF of the two complexes are close and show similar shape distribution.

## 4. Discussion

*S. aureus* is a Gram-positive spherical bacterium. When the body has low resistance or deep wounds, *S. aureus* easily invades and causes infection. Infection with *S. aureus* during hospital surgery easily causes pneumonia, pseudomembranous enteritis, pericarditis, and even systemic infections such as sepsis [27]. However, *S. aureus* is resistant to the currently available antibiotics, so the existing antibiotic drugs cannot meet the treatment needs. MurD ligase is an intracellular continuous enzyme involved in bacterial peptidoglycan biosynthesis. It is an important target for the development of antibiotics. The synthesis of peptidoglycan is divided into several steps, including the assembly of 5′-diphosphate (UDP)—MurNAc pentapeptide in cells, followed by translocation through the plasma membrane and binding to the growing peptidoglycan layer. Peptidoglycan is a unique cell wall component of prokaryotic cells; therefore, there is a growing interest in the use of enzymes involved in the biosynthesis of peptidoglycan precursors [28,29,30].

In recent years, due to the dilemma of land drug research and development, people began to put new drug research and development into marine natural products. More and more target drugs have come from the ocean, and marine drugs are playing an increasingly important role in bacterial inhibition. Many substances extracted from marine organisms have been identified as playing an important role in bacteriostasis, anti-tumor, anti-aging and other aspects. Therefore, Figure 8 shows the workflow of this study. In this study, a virtual screening of the marine small molecule library was initiated to identify a new *S. aureus* inhibitor from the marine small molecule library [30,31]. Through virtual screening, similarity screening and ADME drug analysis, we finally selected 46604 and 46608 as possible drug molecules targeting MurD enzyme of *S. aureus*. Both 46604 and 46608 can have better docking posture in docking, and their docking scores are higher than those of inhibitor D1. The docking score of 46604 is −6.035, compared with −6.243 for 46608. In order to verify the reliability of the docking score, we selected 4 drugs approved by the FDA and the inhibitor D1 selected in this article for precise docking at the same site. The results are as follows. From Table 3, we can see that for the docking of the positive inhibitor D1, the score is −4.900, and the docking scores of the selected drugs are −8.048, −5.200 and −4.684. What is interesting is that, compared with the selected compounds 46604 and 46608, the docking scores of the compounds 46604 and 46608 exceed most inhibitors (*N*-[(6-butoxynaphthalen-2-yl)sulfonyl]-l-glutamic acid, *N*-[(6-butoxynaphthalen-2-yl)sulfonyl]-d-glutamic acid, Lysine Nz-Carboxylic Acid, D1), but smaller than the latest inhibitor N-({6-[(4-CYANO-2-FLUOROBENZYL)OXY]NAPHTHALEN-2-YL}SULFONYL)-D-GLUTAMIC ACID, which to some extent confirms our construction and the accuracy of the model for predicting the docking score. Notably, 46604 has essential hydrophobic interactions with Lys19 and Asn145 residues. Residues Glu166, Ser168, Lys328 and Thr330 tightly bind 46604 to protein in the docking cavity through H-bond. Note that 46608 has strong hydrophobic effect with residues Thr330 and Phe431, residues Asn145, Ser168, and Lys328 formed H-bond interactions with 46608. Through further analysis, it can be seen that both of them form a salt bridge through residue Lys328 (Figure 9A shows the 3D conformation of 46604 in the MurD enzyme protein binding pocket. Figure 9B shows the 2D interaction of 46604 with the protein. Figure 9C shows the 3D conformation of 46608 in the MurD enzyme protein binding pocket. Figure 9D shows the 2D interaction of 46608 with the protein). The abundant interaction types in the two complexes show that the docking results are accurate. In order to further verify the rationality of docking, the binding energies of the top 20 compounds with similarity scores were calculated by Maestro 11.5 MM -GBSA, in which the highest binding energy was −51.3kcal/mol, the lowest was −23.86 kcal/mol, and the binding energies of 46604 and 46608 were −33.74 kcal/mol and −32.48 kcal/mol, respectively (Table 4). Binding energy analysis further shows that they can be stably bound to protein pockets by interaction with proteins to form stable complexes.

In previous studies, the strong binding ability of D1 inhibitors to MurD enzymes was thought to be achieved mainly by the interaction between the complexes. In addition, the hydroxy group consolidates the posture of D1 in the protein pocket by forming hydrogen bonds. Figure 10A shows the pharmacophore model of D1 inhibitors, Figure 10B shows the pharmacophore model based on the 46604 complex, and Figure 10C shows the pharmacophore model based on the 46608-protein complex. The hydroxyl groups on the benzene ring are labeled as hydrogen bond donors while the hydroxyl groups on the carboxylic acid are labeled as hydrogen bond receptors. It is noteworthy that π–π superposition interaction exists in indole rings. The similarity of 46604 and 46608 is 0.447 and 0.445, respectively. The structures of the two benzene rings are reflected in all three compounds, and the structures of the two benzene rings play an important role in all three compounds. Moreover, the hydrogen bond donors and acceptors of the three compounds can be overlapped to some extent in the pharmacodynamic superposition diagram of Figure 10D. The three hydroxyl groups in the 46604 benzene ring also play a strong hydrogen bond with the protein side chain. In addition to hydrogen bonding, hydrophobic interactions also exist on the benzene ring. Like 46604, 46608 has three hydroxyl groups attached to the benzene ring. Hydrogen bond and hydrophobic interaction with protein further confirmed the accuracy of screening results. Indole rings on D1 inhibitors are not reflected in 46604 and 46608, but 46604 and 46608 can form salt bridges with proteins, making docking effect more stable and reliable.

Dynamic simulation results indicate that the complexes of 46604 and 46608 can converge to the equilibrium state in the last 25 ns. It is verified that the two small molecules can exist in a stable conformation in a limited pocket. Detailed analysis of root mean square fluctuation (RMSF) determines residue fluctuation and flexibility throughout the simulation period. The RMSF of two complexes are close and show similar shape distribution. Therefore, the results of molecular dynamics show that 46604 and 46608 have great prospects as inhibitors of *S. aureus*.

## 5. Conclusions

In recent years, food poisoning events caused by *S. aureus* have emerged one after another. Moreover, *S. aureus* has become the second most susceptible bacterium in the United States after *E. coli*. In this study, the homologous model of *S. aureus* was prepared and optimized by MD. Then, three marine molecular libraries CMNPD, MNP and SWMD were used to screen the homologous proteins based on the structure, and the 2010 compounds were screened for similarity. Results showed that 20 compounds were obtained for ADME drug analysis. The calculation of MM-GBSA shows that MM-GBSA dG bind, MM-GBSA dG bind VDW are the contributors to the stable binding of the complex. The most promising compounds 46604 and 46608 in ADME drug evaluation were subjected to 100 ns MD to verify the stability of docking effect, and results showed that both complexes could achieve equilibrium at 75 ns, which confirmed the accuracy of docking effect. In this study, marine small molecules were used for large-scale virtual screening and molecular dynamics simulation of *S. aureus*. In addition, in summary, 46604 and 46608 compounds were proven to be new compounds targeting MurD enzyme of *S. aureus*, which provided a theoretical basis for marine drugs targeting *S. aureus*.

## Figures and Tables

**Figure 1 molecules-26-06426-f001:**
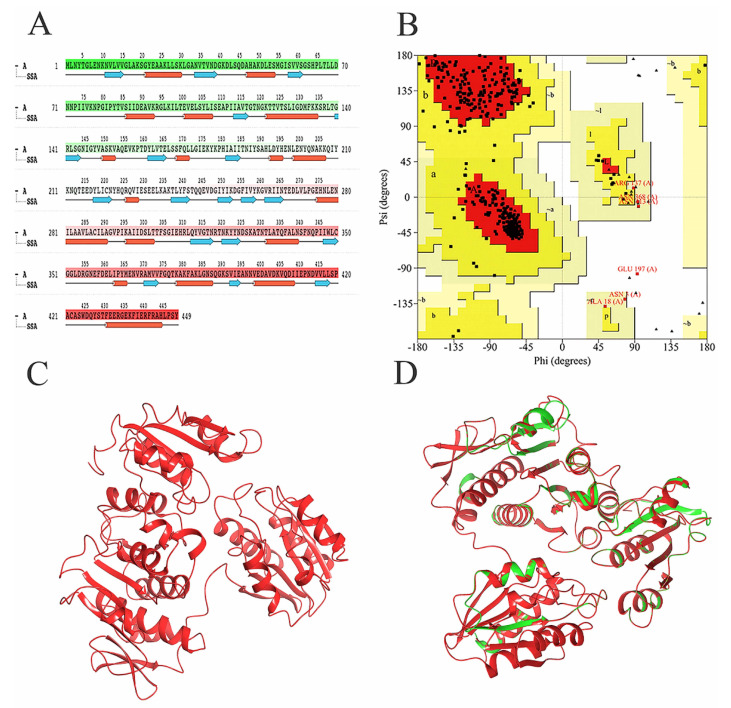
Demonstration and validation of MurD enzyme modeling proteins. (**A**) MurD Ligase sequence of *S. aureus.* (**B**) The Laplace diagram verification of *S. aureus* MurD Ligase represents 358 residues in the most favored region (red), 39 residues in the additional allowable region (yellow), 5 residues in the generous allowable region (light yellow), and 1 residue in the prohibited region (white). (**C**) Homologous model of MurD Ligase of *S. aureus.* (**D**) 3D comparison between MurD Ligase model shown in red and MurD crystal structure of *Streptococcus agalactiae* shown in green (PDB: 3lk7) as template.

**Figure 2 molecules-26-06426-f002:**
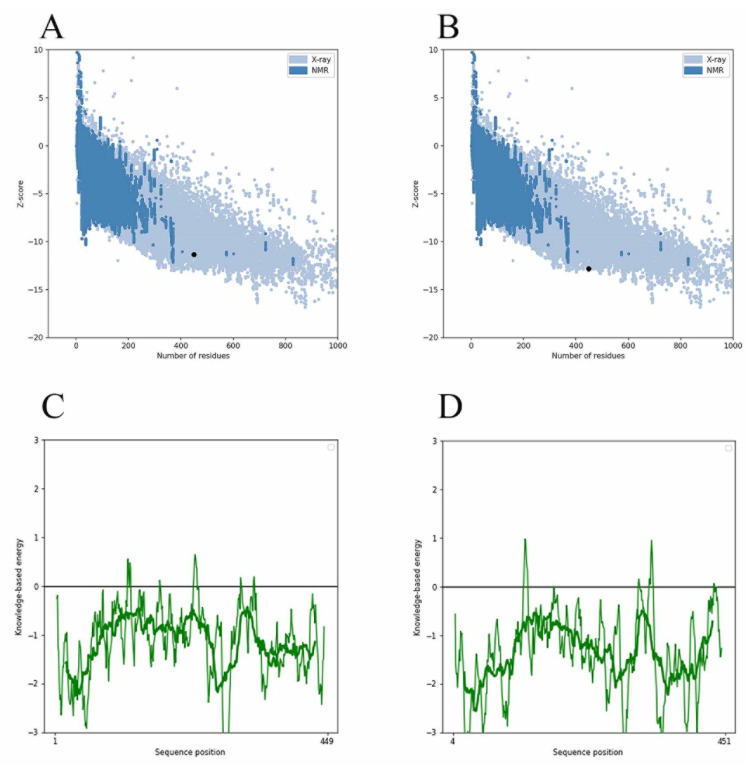
Validation of MurD Ligase model using PROSA. (**A**) MurD enzyme protein overall model quality. (**B**) 3lk7 protein overall model quality; black spots show the similarity of the model with X-ray and NMR structures, X-ray crystallography (light blue region) or NMR spectroscopy (dark blue region). (**C**) MurD enzyme protein residue energy diagram. (**D**) 3lk7 protein residue energy diagram.

**Figure 3 molecules-26-06426-f003:**
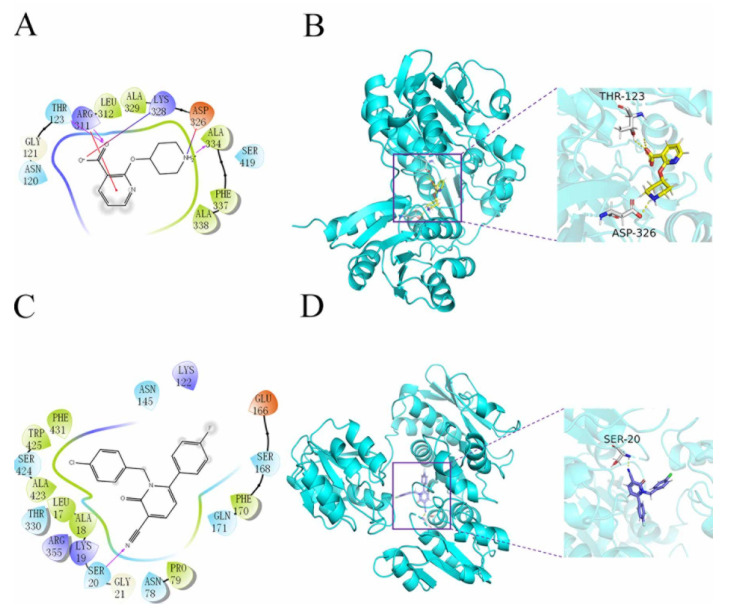
2D and 3D images of 57568 and 3702 interacting with proteins. (**A**) 2D action diagram 57568. (**B**) 3D action diagram of 57568. (**C**) 2D action diagram of 3702. (**D**) 3D action diagram of 3702. The yellow and purple bars represent 57568 and 3702 compounds, respectively. The gray bars represent hydrogen bonding residues, and the yellow dotted lines represent hydrogen bonding.

**Figure 4 molecules-26-06426-f004:**
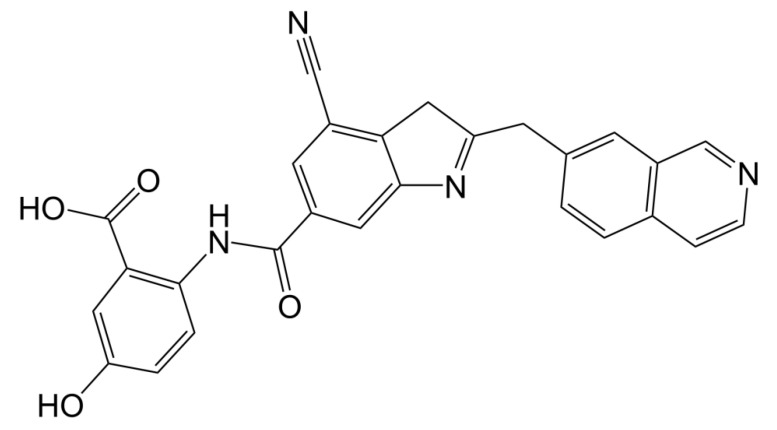
2D structure of D1 inhibitor.

**Figure 5 molecules-26-06426-f005:**
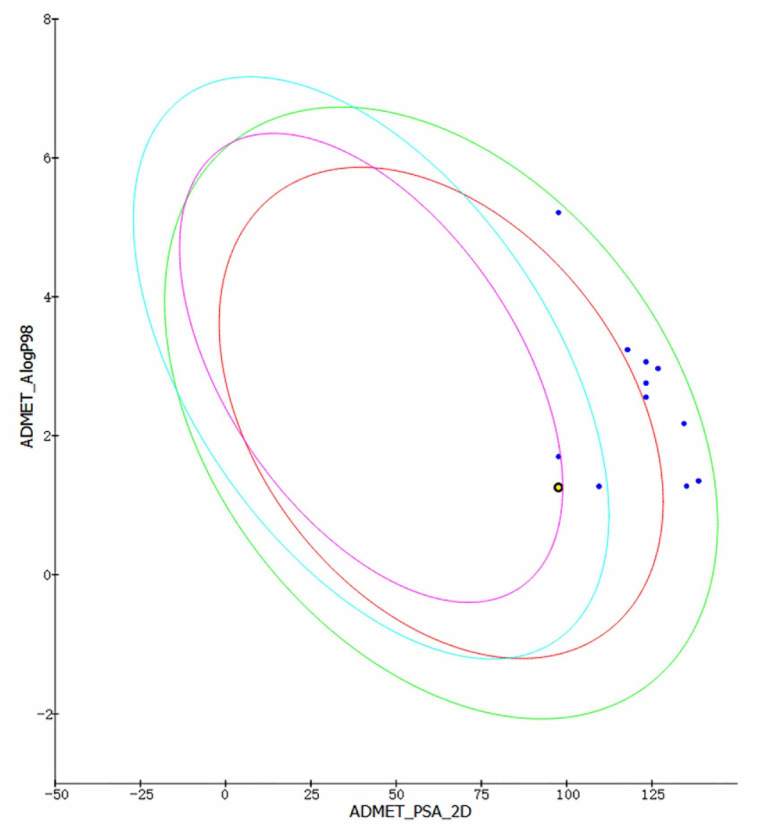
Intestinal absorption model. The purple circle represents 95% of well-absorbed compounds.

**Figure 6 molecules-26-06426-f006:**
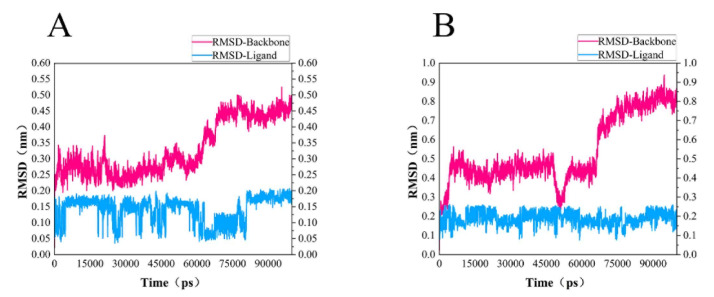
RMSD of protein and ligand. (**A**) RMSD diagram of 46604 and MurD enzyme complex backbone (magenta) and ligand 46604 backbone (blue). (**B**) RMSD diagram of 46608 and MurD enzyme complex backbone (magenta) and ligand 46608 backbone (blue).

**Figure 7 molecules-26-06426-f007:**
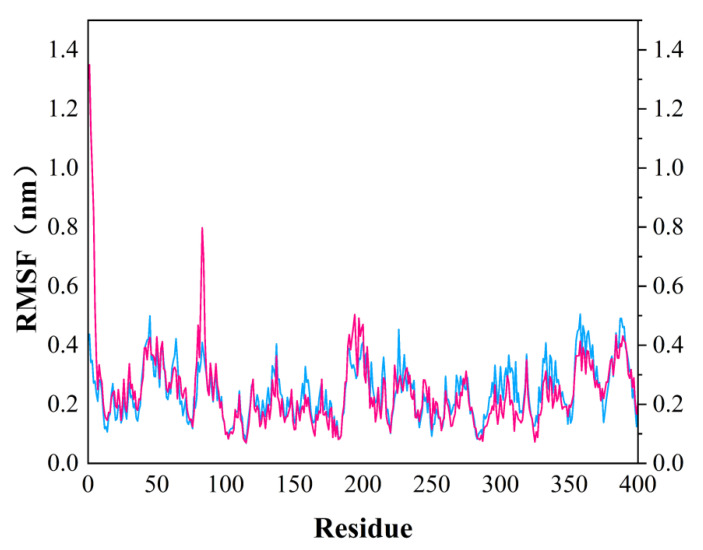
RMSF diagram of MurD Ligase with 46604 (magenta) and 46608 (blue) complexes.

**Figure 8 molecules-26-06426-f008:**
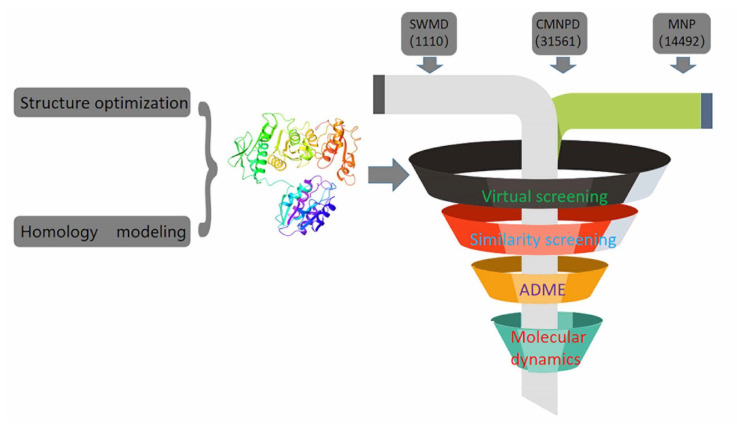
Execution diagram of virtual screening and molecular dynamics.

**Figure 9 molecules-26-06426-f009:**
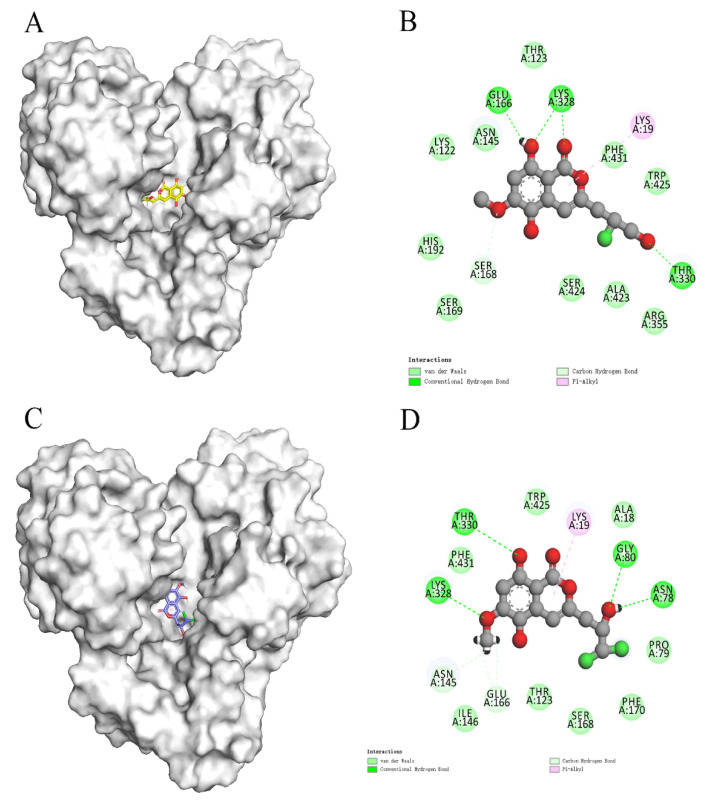
Two-dimensional interaction of 46604 and 46608 with MurD enzyme proteins. (**A**) 46604 (yellow) in the MurD Ligase binding bag. (**B**) 2D display of 46604 interaction in the MurD Ligase binding bag. (**C**) 46608(purple) in the MurD Ligase binding bag. (**D**) 2D display of 46608 interaction in the MurD Ligase binding bag.

**Figure 10 molecules-26-06426-f010:**
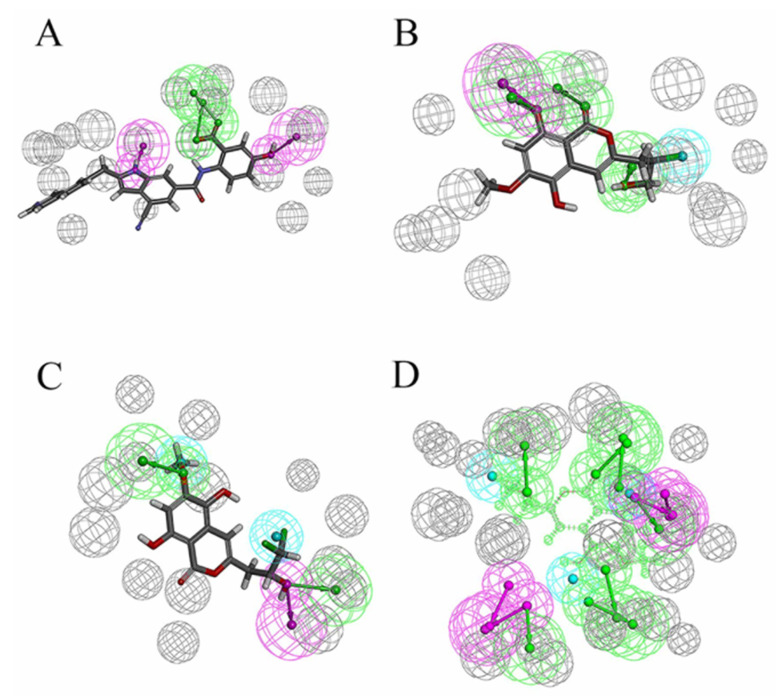
Pharmacophore model and its superposition. (**A**) Pharmacophore model of D1 inhibitor. (**B**) Pharmacophore model of 46604. (**C**) Pharmacophore model of 46608. (**D**) Pharmacophore model superposition. Green spheres represent hydrogen bond receptors, purple spheres represent hydrogen bond donors, blue spheres represent hydrophobicity, and gray spheres represent.

**Table 1 molecules-26-06426-t001:** Binding sites predicted by Sitemap.

	Site Score	Size	D Score	Volume-1
Site 1	1.007	379	1.033	1125.04
Site 2	0.607	34	0.57	82.663
Site 3	0.817	35	0.795	106.673
Site 4	0.644	31	0.545	54.194
Site 5	0.625	31	0.507	102.9

**Table 2 molecules-26-06426-t002:** ADME properties of the top twenty compounds in similarity scoring.

Ligand Name	ADMET Solubility		ADMET PPB Prediction		ADMET AlogP98	ADMET BBB Level	PSA (TPSA)
CMNPD46604	−2.045		TRUE		1.255	3	107.572
CMNPD47366	−3.783		FALSE		2.758	4	151.503
CMNPD47366	−3.783		FALSE		2.758	4	151.502
CMNPD46608	−2.728		TRUE		1.699	3	106.016
CMNPD45974	−2.188		FALSE		1.348	4	154.095
CMNPD45974	−2.188		FALSE		1.348	4	154.095
CMNPD45975	−2.188		FALSE		1.348	4	154.095
CMNPD47365	−3.575		FALSE		2.555	4	152.508
CMNPD47365	−3.575		FALSE		2.555	4	152.508
CMNPD45454	−2.568		FALSE		1.274	4	139.005
MNP45266	−3.689		FALSE		2.175	4	143.298
CMNPD45751	−4.24		FALSE		2.967	4	136.032
CMNPD45751	−4.24		FALSE		2.967	4	136.032
CMNPD46650	−6.014		TRUE		5.213	4	103.449
CMNPD47369	−3.805		FALSE		3.238	4	132.719
CMNPD47369	−3.805		FALSE		3.238	4	132.719
CMNPD45909	−3.886		TRUE		3.064	4	136.694
CMNPD45909	−3.886		TRUE		3.064	4	136.695
CMNPD46636	−2.01		FALSE		1.271	3	119.621
CMNPD46636	−2.01		FALSE		1.271	3	119.621
**Ligand Name**	**QPlogHERG**	**QPPCaco**		**QPlogPo/w**	**QPlogS**	**QPlogBB**	**QPPMDCK**
CMNPD46604	−4.273	168.758		1.381	−2.909	−1.437	136.275
CMNPD47366	−5.068	28.621		1.761	−4.33	−2.444	10.622
CMNPD47366	−5.068	28.621		1.761	−4.33	−2.444	10.622
CMNPD46608	−4.456	220.112		2.006	−3.79	−1.125	467.037
CMNPD45974	−4.746	19.824		0.444	−2.618	−2.594	7.142
CMNPD45974	−4.746	19.824		0.444	−2.618	−2.594	7.142
CMNPD45975	−4.746	19.824		0.444	−2.618	−2.594	7.142
CMNPD47365	−5.199	24.908		1.721	−4.41	−2.547	9.141
CMNPD47365	−5.199	24.908		1.721	−4.41	−2.547	9.141
CMNPD45454	−2.816	17.822		3.132	−4.241	−2.004	8.096
MNP45266	−6.22	24.646		0.813	−4.687	−2.652	9.037
CMNPD45751	−4.554	67.348		1.62	−4.024	−1.775	26.786
CMNPD45751	−4.554	67.348		1.62	−4.024	−1.775	26.786
CMNPD46650	−5.264	176.595		2.775	−4.51	−1.445	75.932
CMNPD47369	−4.592	48.868		1.963	−3.765	−2.029	18.938
CMNPD47369	−4.592	48.868		1.963	−3.765	−2.029	18.938
CMNPD45909	−3.207	14.983		3.405	−5.11	−2.18	6.712
CMNPD45909	−3.206	14.988		3.404	−5.109	−2.18	6.714
CMNPD46636	−4.622	66.658		0.418	−2.415	−1.852	26.49
CMNPD46636	−4.622	66.658		0.418	−2.415	−1.852	26.49

**Table 3 molecules-26-06426-t003:** The docking fraction of compound 46604 and 46608 with the inhibitor.

Molecular ID	Docking Score
CMNPD46604	−6.035
CMNPD46608	−6.243
D1	−4.900
*N*-({6-[(4-CYANO-2-FLUOROBENZYL)OXY]NAPHTHALEN-2-YL}SULFONYL)-d-GLUTAMIC ACID	−8.048
*N*-[(6-butoxynaphthalen-2-yl)sulfonyl]-l-glutamic acid	−5.200
*N*-[(6-butoxynaphthalen-2-yl)sulfonyl]-d-glutamic acid	−5.200
Lysine Nz-Carboxylic Acid	−4.684

**Table 4 molecules-26-06426-t004:** MM-GBSA binding energy of the first 20 compounds with similarity.

Ligands Name	MM-GBSA dG Bind	MM-GBSA dG Bind vdW
MNP45266	−59.25	−34.14
CMNPD45909	−26.02	−37.93
CMNPD45974	−29.93	−30.38
CMNPD147369	−35.93	−35.92
CMNPD47369	−37.87	−35.44
CMNPD45909	−29.36	−26.17
CMNPD45974	−33.15	−26.04
CMNPD45751	−29.19	−23.93
CMNPD45454	−23.86	−31.44
CMNPD46608	−32.48	−21.88
CMNPD47366	−34.05	−38.77
CMNPD47366	−34.7	−39.54
CMNPD46604	−33.74	−30.06
CMNPD47365	−34.52	−34.25
CMNPD47365	−32.5	−38.65
CMNPD46636	−33.39	−20.8
CMNPD46636	−34.15	−22.94
CMNPD46650	−51.3	−29.79
CMNPD45975	−32.27	−30.34
CMNPD45751	−26.97	−31.21

## Data Availability

The data used to support the findings of this study are included within the article.

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
