# Peer review of "Identification of Potential Inhibitors of MurD Enzyme of Staphylococcus aureus from a Marine Natural Product Library"

_molecules, 2021, doi:10.3390/molecules26216426_

Round 1

Reviewer 1 Report

In the manuscript “Identification of Potential Inhibitors of MurD Enzyme of Staphylococcus Aureus from Marine Natural Product Library”, the authors did virtual screen of three Marine natural product Libraries and found that two compounds CMNPD46604 and CMNPD46608, as inhibitors, to be potential drug candidates against staphylococcus aureus. The compound characterization was demonstrated with good performing physicochemical properties (ADME), efficacy of the docking effect (computational fluid dynamics), and tightly binding affinity.

Due to many antibiotics are ineffective in treating Staphylococcus aureus infections, the two compounds might be lead to new drugs for therapy of the people infected with the bacteria.

Two minor comment

  • IT is good to show the 2D structure of CMNPD46604 and CMNPD46608.
  • Line 402: Figure 9. A) Super-position of 46604 (yellow) and 46608 (purple) in the binding pocket of the main protease.

         It is hard to see the difference between yellow and purple.

Author Response

Reviewer #1:

In the manuscript Identification of Potential Inhibitors of MurD Enzyme of Staphylococcus

Aureus from Marine Natural Product Library, the authors did virtual screen of three Marine

natural product Libraries and found that two compounds CMNPD46604 and CMNPD46608, as

inhibitors, to be potential drug candidates against staphylococcus aureus. The compound

characterization was demonstrated with good performing physicochemical properties (ADME),

efficacy of the docking effect (computational fluid dynamics), and tightly binding affinity.

Due to many antibiotics are ineffective in treating Staphylococcus aureus infections, the two

compounds might be lead to new drugs for therapy of the people infected with the bacteria.

  1. IT is good to show the 2D structure of CMNPD46604 and CMNPD46608.

ResponseThank you for the constructive comments. We have replaced the 2D drawings of Figure 9. CMNPD46604 and CMNPD46608, as shown below.

  1. Line 402: Figure 9. A) Super-position of 46604 (yellow) and 46608 (purple) in the binding pocket of the main protease. It is hard to see the difference between yellow and purple.

ResponseThank you for the comments by the reviewer. We noticed the overlapping position of the two compounds in Figure 9 and replaced the picture, as shown below.

Figure 9. Two-dimensional interaction of 46604 and 46608 with MurD enzyme proteins.(A) 46604(yellow) in the Murd Ligase binding bag (B) 2D display of 46604 interaction in the Murd Ligase binding bag (C) 46608(purple) in the Murd Ligase binding bag (D) 2D display of 46608 interaction in the Murd Ligase binding bag.

Reviewer 2 Report

Authors describe identifying potential inhibitors of MurD enzyme of Staphylococcus aureus from marine natural product library by in silico screening. S. aureus is a common pathogen in human suppurative infection. Antibiotics work in general, however, the recent emergence of resistant strains has been a challenging problem. Therefore, the results in this manuscript could contribute to developing novel drugs against the bacteria. 
The manuscript is well prepared, and the data and results are also scientifically sound. Therefore, this submission could be considered for publication with some revisions as follows.

Bacterial strain names should be in italic, and from the second appearance, the name should be abbreviated such as S. aureus and E. coli.

There are crystal structures complexed with inhibitors in Protein Data Bank. Authors should compare and discuss the binding modes of compounds in this study with those in PDB.
With the above, the binding site of the compound screened in this study covers the active site and/or ATP binding site? These discussions would add some biological significance by in silico experiments.

Line 243, protease ? Ligase?

Line 296, Reference should be inserted.

Line 361, sepsis and sepsis (duplicated?)

Author Response

Reviewer #2:

Authors describe identifying potential inhibitors of MurD enzyme of Staphylococcus aureus from marine natural product library by in silico screening. S. aureus is a common pathogen in human suppurative infection. Antibiotics work in general, however, the recent emergence of resistant strains has been a challenging problem. Therefore, the results in this manuscript could contribute to developing novel drugs against the bacteria. The manuscript is well prepared, and the data and results are also scientifically sound. Therefore, this submission could be considered for publication with some revisions as follows.

1. Bacterial strain names should be in italic, and from the second appearance, the name should be abbreviated such as S. aureus and E. coli.

ResponseThank you for your valuable comments. We have changed the strain names to italics and changed the reappearing strain names below to the abbreviation S. aureus and marked them yellow in this article.

Staphylococcus aureus

  1. aureus

2.There are crystal structures complexed with inhibitors in Protein Data Bank. Authors should compare and discuss the binding modes of compounds in this study with those in PDB. With the above, the binding site of the compound screened in this study covers the active site and/or ATP binding site? These discussions would add some biological significance by in silico experiments.

ResponseThank you for your comments. It has been shown that the ATP binding site of MurD is located in the gap between the C-terminal and the central domain. The central domain contains a single nucleotide binding fold responsible for the binding of adenosine triphosphate (ATP)[1][2][3]. According to the analysis of the docking results in this paper, the predicted pocket is in the central structural domain. In the literature, Lys19, Gly147, Tyr148, Lys328, Thr330, and Phe431 residues are responsible for the stabilization of the inhibitor - protein complex by binding 2-Thiox-Othiazolidin-4-One to MurD ligase[4].In this study, the docking compound also established a stable connection relationship through some residues in the pocket, which further verified the prediction that the pocket was located at the ATP binding site. In addition, Sitemap in Schrodinger has been verified by several studies to have the ability to accurately predict pockets, ensuring that the prediction of MurD ligase docking pocket in this paper is accurate, and the inhibitor study is sufficient for further experimental verification[5].

[1] Perdih A, Kotnik M, Hodoscek M, Solmajer T. Targeted molecular dynamics simulation studies of binding and conformational changes in E. coli MurD. Proteins. 2007 Jul 1;68(1):243-54. doi: 10.1002/prot.21374. PMID: 17427948.

[2] Bertrand JA, Auger G, Martin L, Fanchon E, Blanot D, Le Beller D, van Heijenoort J, Dideberg O. Determination of the MurD mechanism through crystallographic analysis of enzyme complexes. J Mol Biol. 1999 Jun 11;289(3):579-90. doi: 10.1006/jmbi.1999.2800. PMID: 10356330.

[3] Gordon E, Flouret B, Chantalat L, van Heijenoort J, Mengin-Lecreulx D, Dideberg O. Crystal structure of UDP-N-acetylmuramoyl-L-alanyl-D-glutamate: meso-diaminopimelate ligase from Escherichia coli. J Biol Chem. 2001 Apr 6;276(14):10999-1006. doi: 10.1074/jbc.M009835200. Epub 2000 Dec 20. PMID: 11124264.

[4] Azam MA, Jupudi S, Saha N, Paul RK. Combining molecular docking and molecular dynamics studies for modelling Staphylococcus aureus MurD inhibitory activity. SAR QSAR Environ Res. 2019 Jan;30(1):1-20. doi: 10.1080/1062936X.2018.1539034. Epub 2018 Nov 8. PMID: 30406684.

[5] Halgren T. New method for fast and accurate binding-site identification and analysis. Chem Biol Drug Des. 2007 Feb;69(2):146-8. doi: 10.1111/j.1747-0285.2007.00483.x. PMID: 17381729.

3.  Line 243, protease ? Ligase?

ResponseThank you for the comments. We have replaced the inappropriate use of protease in the article with Ligase, please see the highlighted yellow part of the article.

4.  Line 296, Reference should be inserted. ResponseThank you for the insightful comments by the reviewer. We have added the literature in line 296, as highlighted in yellow below [ ] Sastry GM, Dixon SL, Sherman W. Rapid shape-based ligand alignment and virtual screening method based on atom/feature-pair similarities and volume overlap scoring. J Chem Inf Model. 2011 Oct 24;51(10):2455-66. doi: 10.1021/ci2002704. Epub 2011 Sep 15. PMID: 21870862.

  1. Line 361, sepsis and sepsis (duplicated?)

ResponseThank you for the constructive comments. We have cut out the repetition

Reviewer 3 Report

The manuscript reports the application of large array of computational methods for the identification of new MURd inhibitors. The manuscript is not well written, many sentences are unclear and also the use of references is not suitable. It was very difficult to go thought the manuscript. The main concern is about the absence of validation. What about the already published inhibitors? Is the model able to correctly scores the known inhibitors?. I do not think that the manuscript fit the journal quality requirements, thus i do not suggest it for publication  

Author Response

Reviewer #3:

The manuscript reports the application of large array of computational methods for the identification of new MurD inhibitors.

  1. The manuscript is not well written, many sentences are Unclear.

ResponseThank you for the insightful comments. We have revised the unclear sentences in the article and polished the manuscript. The revised parts are highlighted in yellow in the manuscript.

  1. Also the use of references is not suitable.

ResponseThank you for the insightful comments by the reviewer. We have deleted or changed the references improperly cited.

  1. The main concern is about the absence of validation. What about the already

published inhibitors? Is the model able to correctly scores the known inhibitors?

ResponseThanks to reviewers for their insightful comments. In view of the rapid emergence and spread of drug-resistant bacteria such as Gram-positive bacteria staphylococcus aureus, most of the drugs that have been put into use have proved ineffective against Staphylococcus aureus, so we did not directly choose to study existing inhibitors [1].Instead, we further designed a new inhibitor D1 based on existing drugs and literature[2].D1 can be further confirmed by the results of drug molecular dynamics studies. In order to verify the accuracy of site and Model scores, docking scores of FDA-approved drugs and MurD ligase were supplemented in the discussion section. FDA-approved drugs maintain stable conformation by docking in the central region of the MurD ligase and form stable protein ligand interactions through hydrophobic interactions. Since there are only a dozen Marine drugs on the market, far fewer than terrestrial natural products, the purpose of this paper is to explore effective drugs against drug-resistant bacteria staphylococcus aureus from Marine natural products. We look forward to using experiments to verify the screening results in future studies. Additions to the discussion section are shown in bold yellow.

[1] Nambiar S, Laessig K, Toerner J, Farley J, Cox E. Antibacterial drug development: challenges, recent developments, and future considerations. Clin Pharmacol Ther. 2014 Aug;96(2):147-9. doi: 10.1038/clpt.2014.116. PMID: 25056394.

[2] Azam MA, Jupudi S, Saha N, Paul RK. Combining molecular docking and molecular dynamics studies for modelling Staphylococcus aureus MurD inhibitory activity. SAR QSAR Environ Res. 2019 Jan;30(1):1-20. doi: 10.1080/1062936X.2018.1539034. Epub 2018 Nov 8. PMID: 30406684.

In order to verify the reliability of docking score, four drugs approved for marketing by FDA were selected for precise docking with the inhibitor D1 selected in this paper at the same site. The results are shown in Table 3. It can be seen from the table that the docking score of positive inhibitor D1 is -4.900, while the docking score of selected drugs is -8.048 respectively. Interestingly, compared with the screened compound 46604, 46608, compound 46604,46608 docking score more than most of the inhibitors (N - [(6 - butoxynaphthalen - 2 - yl) sulfonyl] - L - glutamic acid, N - [(6 - butoxynaphthalen - 2 - yl) sulfonyl] - D - glutamic acid, Lysine Nz - Carboxylic acid, D1),However, it is smaller than the latest inhibitor N-({6-[(4-cyano-2-fluorobenzyl)OXY] Naphthalen-2-yl}SULFONYL) d-glutamic ACID, which confirms the accuracy of our model for predicting butt score to a certain extent.

Table 3.The docking fraction of compound 46604 46608 with the inhibitor

Molecular ID                                                                                    

Docking

Score

CMNPD46604

-6.035

CMNPD46608

-6.243

D1

-4.900

N-({6-[(4-CYANO-2-FLUOROBENZYL)OXY]NAPHTHALEN-2-YL}SULFONYL)-D-GLUTAMIC ACID

-8.048

N-[(6-butoxynaphthalen-2-yl)sulfonyl]-L-glutamic acid

 -5.200

N-[(6-butoxynaphthalen-2-yl)sulfonyl]-D-glutamic acid

 -5.200

Lysine Nz-Carboxylic Acid

 -4.684

Reviewer 4 Report

This manuscript, "Identification of potential inhibitors of MurD enzyme of Staphylococcus aureus...," by Zheng et al. is interesting insofar as it goes. This is an entirely in silico investigation, and gives what they describe as "possible drug molecules" (page 12; line 380) or "drug candidates" (abstract; line 12). At this early stage of investigation, the identified molecules (46604 and 46608) can surely be referred to as leads, if that. So the manuscript needs to be refocused to reflect this more realistic situation.

Also, there are a few minor adjustments that need to be addressed:

Line 33: "layer by layer" is an odd statement.

Line 43: "sepsis and sepsis". Use only once here.

Line 46: "infectious body" is not a typical usage.

In the Materials and Methods section, the present tense is incorrectly used; this should be changed to past tense, because the authors are describing what they did.

Otherwise, the manuscript is well-written and describes a body of work that is carried out apparently well.

Author Response

Reviewer #4:This manuscript, "Identification of potential inhibitors of MurD enzyme of Staphylococcus aureus...," by Zheng et al. is interesting insofar as it goes. 1.  This is an entirely in silico investigation, and gives what they describe as "possible drug molecules" (page 12; line 380) or "drug candidates" (abstract; line 12). At this early stage of investigation, the identified molecules (46604 and 46608) can surely be referred to as leads, if that. So the manuscript needs to be refocused to reflect this more realistic situation.

ResponseThank you for your comments. In this study, some compounds with good scores were obtained through a wide range of virtual screening, and the most promising 46604 and 46608 were obtained by similarity screening and ADME drug-forming screening. At last, 100 ns molecular dynamics simulation was performed to verify the two inhibitor complexes. The results show that both of them can dock with protein and have good docking effect, forming stable protein-ligand interaction. At the same time, given the scarcity of Marine natural product drugs and the resistance of S. aureus to drugs already developed on land, the results obtained in this paper are likely to be a novel inhibitor for the treatment of S. aureus.

2.  Line 33: "layer by layer" is an odd statement.

ResponseThank you for the valuable comments. We have replaced the expression "layer by layer" as follows:

Bacterial infections have become a common case worldwide and the development of antibiotics has been hampered by bacterial resistance.

3.  Line 43: "sepsis and sepsis". Use only once here.

ResponseThank you for your valuable suggestion. We have deleted the duplicated parts.

4.  Line 46: "infectious body" is not a typical usage.ResponseThank you for the comments by the reviewer. We've deleted the word "infectious body". 5.  In the Materials and Methods section, the present tense is incorrectly used; this should be changed to past tense, because the authors are describing what they did.

ResponseThank you for the insightful comments by the reviewer. We have corrected improper tenses in the Materials and Methods section.

Round 2

Reviewer 3 Report

The revised form of the manuscript is of higher quality. Some spell check are still required. Just one point has to be fixed, how do the author justify the different binding mode of the two selected compounds. The structures of the  selected compounds are very similar but in the picture 9 panel A and C the binding mode seems different.

Minor point: page 2 row 96 Material and Method the resolution of the crystal structure is in Angstrom

Author Response

Reviewer #3:

  • The revised form of the manuscript is of higher quality. Some spell check are still required.

ResponseThank you for the valuable comments by the reviewer. We apologize for the grammatical and formatting errors in the manuscript, and we have revised the manuscript and supplementary materials accordingly carefully.

  • Just one point has to be fixed, how do the author justify the different binding mode of the two selected compounds. The structures of the selected compounds are very similar but in the picture 9 panel A and C the binding mode seems different.

ResponseThank you for your comments. The structurally similar CMNPD 46604 and CMNPD 46608 have different binding patterns in binding pockets, which are mainly affected by pocket shape and compound structure. In this experiment, the docking pocket is located in the central structural domain of ATP docking, and the pocket space is large, while the two compounds are relatively small, so it is normal to have different docking postures in the pocket. In addition, the two compounds are also different in structure. In the terminal groups, there are differences between them, mainly reflected in the positions of chlorine atoms and hydroxyl groups, which can also affect the docking results. The influence of groups on the docking results can be seen from the 2D interaction diagram of the two compounds in Figure 9, which shows that compounds are different in the selection of protein residues. Considering the accuracy of the docking posture of the two compounds, molecular dynamics and docking result analysis were used to verify them, and the results can well prove that the position of the two compounds in the pocket is stable and they have a good docking posture.

  • Minor point: page 2 row 96 Material and Method the resolution of the crystal structure is in Angstrom.

ResponseThank you for your comments. We have modified the resolution unit, and the modification has been marked yellow.
